# Product Model Derivation from Feature Model and Formal Specification

**Xi Wang** [1,2], **Weiwei Wang** [1,2,*] and **Hongbo Liu** [1,2]

1    School of Computer Technology and Science, Shanghai University, Shanghai 200444, China;
wangxi@t.shu.edu.cn (X.W.); liuhongbo@shu.edu.cn (H.L.)
2    Shanghai Key Laboratory of Computer Software Testing and Evaluating, Shanghai 201114, China
*    Correspondence: wweiwei@shu.edu.cn

**Abstract:** Product derivation is the process of building a specific product from a software product line. Effective product derivation can improve software reuse productivity. Existing methods can only obtain abstract feature models, lacking detailed specifications of individual features. They are more about deriving code assets or class diagram templates without precise model descriptions for specific products. This article proposes a product derivation approach to obtain a formal specification of a specific product based on the feature model and formal specification. We use the integration ordering and behavior preserving integration techniques to integrate the formal specification for each feature pair. The method is divided into two steps. First, it determines the feature formal specification integration ordering based on the feature model. Then, the behavior-preserving integration will be conducted for pairs, including declaration integration, functional scenario path generation, and function integration based on path matching. Behavior preserving integration guarantees consistent behavior to ensure the quality of the formal specification after integration. Finally, we developed a support tool to conduct a case study. The tool first guides the user to perform feature functional scenario path matching, then performs functional integration based on the matching results and repeats the above steps to generate a product model. The result indicates that our method facilitates the derivation process and improves the quality of the generated models.

**Keywords:** product derivation; formal specification; product family modeling; behavior preserving integration

## 1. Introduction

Software product line development is an important way to realize software reuse, which can improve software quality and shorten the development cycles, time, and cost [1,2]. A product family is a set of software products sharing common features but containing variation points. Deriving a specific product from a product family is called product derivation, which is a crucial activity in software product line development. In a product line organization, using the appropriate product derivation process helps to ensure the required return on investment for developing platform assets [3].

At present, in the study of product derivation, many studies focus on the realization of the variability of product family members at the implementation level [4,5]. The techniques are code-based export methods that assign features to code [6]. They are usually feature-to-artifact mappings [7], e.g., AspectJ for Java or preprocessor annotations [8,9] for text files. The build result of the final product depends on the choice of configuration options provided by the variability model [10]. These studies focus on the ability of a specific type to track changes. However, this approach leads to two problems: (i) The built product has no formal description of features. It is abstract in most cases, and only contains configuration information of variable product features and dependencies between features, emphasizes the feature architecture and relationship analysis, and cannot fully identify and understand

individual features. (ii) There is no systematic modeling method for the product families, resulting in the lack of organically unified variability and feature descriptions.

Compared to the implementation level, some work and tools have been conducted on the requirement level. Most of the existing methods on the requirement level are *copy-based* and *transformation-based*. In the former, an existing product variant is copied and modified, adding and/or removing some functionalities to derive a new product [11,12]. The latter involves transforming feature elements into different forms of the element [13]. Most of them either lack formalization of the model or have a coarse and abstract granularity of formalization, resulting in poor quality products. The major reason is that the source of product derivation is the product family model, which insufficiently expresses the user requirement.

This paper builds on our prior work. Our prior work provided an evolution method for high-quality product family modeling that considers the feature model and the relationship between features, and a consistent and accurate product family model can be obtained [14,15]. It solved the problem that the source model was not formal and provided the basis for the self-service product model generation. This paper significantly extends our prior work to solve the problem that the specific product derived is not formalized or the formalization is imprecise and obtains the formal specification of the product.

Based on the above work, this paper proposes a method for product derivation at the requirements level that integrates the feature specification, which focuses on feature specification rather than code implementation. Our approach has two steps. It starts with determining the feature specification integration ordering based on the feature model. This ordering is the sequence of features to be integrated. The two features to be integrated in order in the sequence are called a feature pair (an atomic integration). Then, for each feature pair, behavior-preserving integration will be conducted according to the ordering, including declaration integration, functional scenario path generation, and function integration based on path matching. Behavior preserving integration combines structural and semantic information present in the feature specification, and the functional scenario path covers all functional scenarios in the individual feature specification. For example, features A and B form a feature pair. The matching of the functional scenario path of pairs will be carried out by analyzing their behaviors. A set of feature integration rules are given for guiding the construction of integrated features. The final product can be obtained when the behavior of the feature pairs to be integrated is preserved in the combined feature. At the same time, since the introduction of the formal specification makes the generated product formal, and the integration process can be supported automatically, we developed a tool based on the method and conducted a case study to illustrate the validity of the approach (the tool can be downloaded and run at https://github.com/Jennywww/PD-SOFL-tool, accessed on 14 May 2022). In summary, we make the following contributions:

- A product model derivation method is proposed, which enables the computer-aided construction of formal product models from a feature model and feature specification;
- A supporting tool is provided to guide through the process of product model construction and ease the burden of product derivation;
- A functional-scenario-based behavior-preserving mechanism is given to guarantee the completeness of the resultant product model.

The underlying proposal is language-independent, so it can be supported by formal specification languages, such as the Z and B-Method. However, we use the Structured Object-Oriented Formal Language (SOFL) because it is a formal engineering method that combines graphics, formal expression [16], and a three-step approach [17], solving the gap between the formal method and the real development of software [18–20]. It is easier to use and more easily accepted by the industry [21].

The remainder of the paper is organized as follows. Section 2 reviews related work. Section 3 introduces the method of product derivation with feature model and formal specification. Section 4 illustrates the validity of the proposed approach through the

supporting tool. Section 5 discusses the advantages and limitations. Finally, in Section 6 we conclude the paper and point out future research directions.

## 2. Related Work

Product derivation is more concerned with the realization of the variability at the implementation level [4]. Souza et al. [22] use a preprocessing-based code export method (GenArch) for product derivation. In [23], the authors use Java to implement the selection of basic components and features from the software product family and derived products. Feichtinger et al. [24] summarize complex relationships between features by computing a feature dependency matrix and use a static code analysis and change control systems to promote complex code-level dependencies to feature models to guide model evolution. Tërnava et al. [25] propose a framework for managing the imperfect modularity of variable implementations. Capturing variability based on feature change points, then modeling the variability with variants while maintaining consistency with implementations in code assets and establishing tracking links between specified variables and implemented variables, completes the management of implemented variability. Marah et al. [26] propose a model-driven round-trip engineering (RTE) approach where different configurations can be obtained. Both the applications and synchronization between instance models and corresponding codes are provided using a toolchain.

These methods discussed above are limited to the code unit of single features and code of feature interactions. The fine-grained product stays on the feature, and they do not involve the expression of the internal behavior of the feature. In addition, these approaches focus on feature modeling with little consideration for dependencies and interactions between implementation artifacts. In contrast, our method systematically considers feature architecture and combines features into a whole while being able to express the internal behavior of features. At the same time, the specific product features derived from the derivation have dependencies, and the behavior of the product can be retained after the derivation through the behavior preserved integration.

There are also some works on the requirement level for product derivation. Sepúlveda et al. [27] take the captured variability information, such as features, and build a trace between the features and the use case model to the model variability in use cases. Hajri et al. [28] propose a way to support evolving configuration decisions in product line use case models, using use case models to minimize manual tracking efforts, and reconfigure use case diagrams and specifications to accommodate evolving decisions. Some approaches [29,30] use propositional logic for a product derivation to produce Unified Modeling Language (UML) class diagrams or sequence diagrams. Some express the SPL source model in UML and derive the UML model of a specific product by mapping the features in the feature model and their implementation in the design model [13]. Domain-specific modeling languages are defined to automate product derivation [31]. For example, Vještica et al. [32] propose a multi-level production process modeling language (Multi-ProLan) that allows process engineers, quality engineers, and plant managers to collaborate on specifying a production process by using a common language. Nieke et al. [33] proposed augmenting the metamodel with a seamless support for planning, tracking, and slicing model evolution timelines, enabling the arbitrary modeling notations for integrated storage, tracking, planning, and access and control of the model evolution.

The above methods stay at the feature level and are described with the granularity of features. Although some methods have been given for the detailed specification of individual features based on the feature model, most of them are informal and fail to manage the accurate description of feature requirements, making them unable to automate support, which brings difficulties to the later coding, testing, and maintenance of the product. By contrast, our approach performs feature integration by matching functional scenario paths and formal specifications to ensure a diverse interaction between feature behaviors.

Our approach uses matching and integration to fuse feature specifications. Matching and integration have had much prior work in the formalization field. Shiva Nejati [34] proposed a method of matching and merging state diagram models. It uses natural language processing methods to judge the similarity between the semantics of the input models and calculates the matching state pairs with accuracy. Engineers use scenario-based modeling methods to explain behavioral interactions in the early design process. In Ref. [35], the authors discussed the use of Live Sequence Charts (LSCs) [36] to define a scenario specification language and describe possible scenarios through detailed behavior models. In Ref. [37], the authors propose a Scenario-Based Product Line Specification (SBPLS) framework, which combines a feature model with Modal Sequence Diagrams (MSDs) [38], and allows engineers to formally specify interactions in product lines of open reactive systems. Michal Smialek and his team considered extracting code from scenarios and facilitated the transition from the requirements to code [39]. The above methods focus on the possible interactions between features, including real-time sequence diagrams, state diagrams, etc. Most of these methods are mainly based on state transition models, and it is challenging to complete the matching and integration for data-intensive systems. Moreover, these methods do not provide rich data types for defining data structures with our use appropriate data types by Structured Object-Oriented Formal Language (SOFL) to describe feature behaviors.

## 3. Product Derivation Based on Feature Model and Specification Approach

Feature models and formal specifications are the foundation of our article. We use feature models to capture the commonality and variability of features and use SOFL to describe formal specifications of the features. Then, we illustrate the detailed process of the product derivation from the feature model to obtain specific products.

### 3.1. Feature Model

The feature model uses a tree-like hierarchical structure to describe the relationship between features, reflecting the commonalities and variability of features. In order to illustrate the feature model, the Tourism Management System (TMS) is given as an example, which is used in the following sections. The main function of TMS is to customize personalized travel plans according to user needs and complete necessary transactions. The feature model is shown in Figure 1.

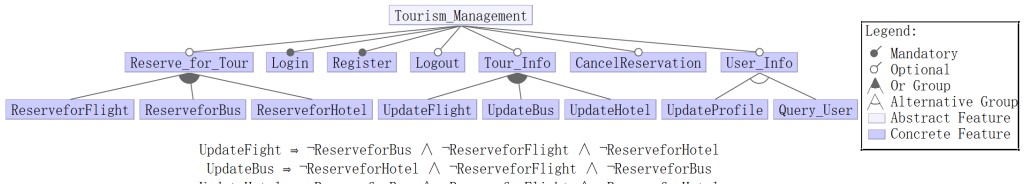

**Figure 1.** The feature model of Tourism Management.

The root node, in this case, is called *Tourism Management*, which can be decomposed into multiple sub-features. A *mandatory* relationship dictates that children are necessary. For example, each product must have *login* and *registration* functions. An *optional* relationship means that a child node can be selected or unselected. For example, the *Reserve_for_Tour* and *Tour_Info* can be included in any product. An *alternative* relationship represents that one of the sub-features must be selected. For example, *UpdateProfile* and *Query_User* cannot appear in the same product. The *Or* relationship representation can contain one or more sub-features. For example, *ReserveforFlight*, *ReserveforBus*, *ReserveforHotel*, can be included in one or more products simultaneously. The three Boolean logic expressions at the bottom of the figure indicate the constraints between features. For example, the logic expression $UpdateFlight \Rightarrow \neg ReserveforBus \cap \neg Reservefor$

*Flight* ∩ ¬*ReserveforHotel* means that if *UpdateFlight* is selected, then *ReserveforFlight*, *ReserveforBus* and *ReserveforHotel* cannot be selected.

### *3.2. SOFL*

SOFL, standing for Structured Object-Oriented Formal Language, is a formal engineering method. A SOFL specification is usually composed of modules that are associated with CDFDs (Condition Data Flow Diagrams). CDFDs are designed in a hierarchy to describe the architecture of the system under design, while the components are used in each CDFD, such as data flows, data stores, and processes. Each process is described in preconditions and postconditions, where the preconditions can be used to describe constraints on input, and postconditions can be used to describe functions on output data. Different processes are linked to each other by data [40]. Figure 2 shows the basic structure of the formal specification to reflect the relationship between module, CDFD, and process. For example, module A1 consists of A11 and A12, CDFD A1 represents the behavior of module A1, and module A1 encapsulates the data in CDFD A1 (const; type; class; var; inv;) and process (A11, A12), which together form the specification of SOFL.

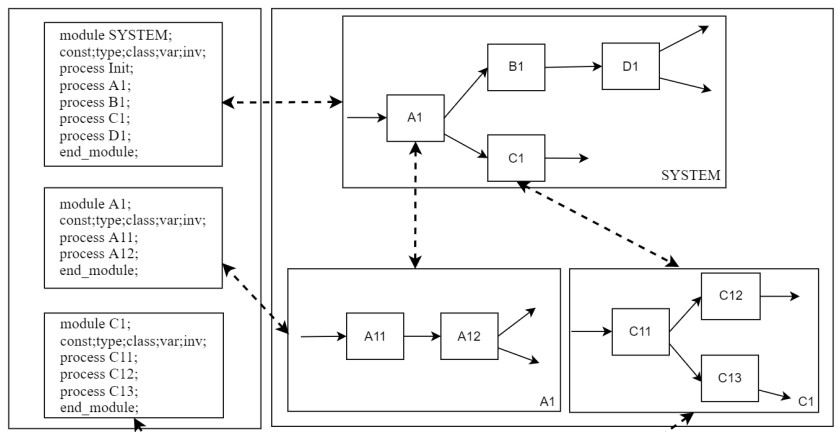

**Figure 2.** The framework of a SOFL specification.

A process in SOFL is the most basic unit that constitutes a module, and a module is formed by integrating a group of processes. Each process is responsible for performing an action, task, or operation, it accepts input and produces output, and different processes are connected through data. Figure 3 is the basic structure diagram of a *Check_Pass* process and the corresponding formal specification. To explain the basic concepts involved in our discussion, we have simplified the original version of the process specification to the extent that we believe ensures a good understandability. A process consists of five parts: name, input ports, output ports, preconditions, and postconditions. In the middle of the box is the name of the process. The input port on the left side of the graphical representation receives three input data flow *sel* (customer selection), *pass_no* (the password provided by the customer to access his account), *id* (the account number provided by the customer), while the three on the right output ports are used to connect different output data flow, allowing users to enter the next stage to perform different operations. For example, the *Check_Pass* process is intended to verify the customer's account and password through the database for further operations, and the output data flows of different ports cannot be valid at the same time. Different outputs are obtained according to different conditions. Details of the conditions are given in the formal specification of the process.

The reader who wishes to understand more details of SOFL can refer to [40] for extensive reading. The formal definitions of the key elements in SOFL are given.

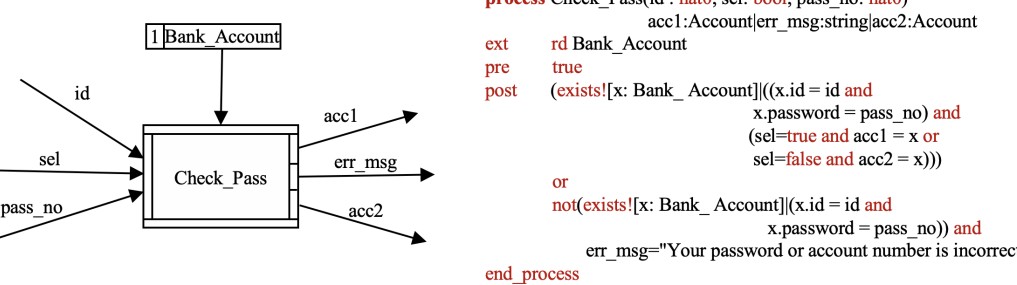

**Figure 3.** A process for checking password.

**Definition 1.** *A SOFL process p is a 4-tuple* $(IPort, OPort, preP, postP)$, *where* $IPort = \{iPort_1, ...iPort_m\}$ *is the set of input ports of p, where each* $iPort_i = \{iv_{i1}, ..., iv_{il}\}$ *is a set of input variables,* $OPort = \{oPort_1...oPort_n\}$ *is the set of output ports of p where each* $oPort_j = \{ov_{j1}, ..., ov_{jk}\}$ *is a set of output variables. The preP is a constraint on the input, and postP is the relationship between input and output. Each preP and postP exists as a union of multiple predicate expressions.*

A process $P = (IPort, OPort, preP, postP)$ as $P = \{P_{ij} | 1 \leq i \leq m, 1 \leq j \leq n\}$, $P_{ij} = (iPort_i, oPort_j, preP_{ij}, postP_{ij})$, where $preP_{ij} = preP$, $postP_{ij} = postP$. The execution of the process consumes all the input data flows $iv_{i1}, ..., iv_{il}$ connected to the available input port $iPort_i$ for activating the execution, and makes exactly one of the output ports $postP_{ij}$ available. Therefore, all the output data flows $ov_{j1}, ..., ov_{jk}$ connected to $oPort_j$ become available. This part will become clearer as the discussion on the functional scenario path of a feature and its specification progresses.

**Definition 2.** *A module m is a tuple* $(P, L, D, C, T, F, \Phi, \lambda)$ *where P is a set of processes, L is a set of labels. D is a set of datastores, T denotes the objects outside the system, C is the set of lower-level CDFDs for decomposing processes in m,* $F : ((OPort(p) \cup T) \times L) \rightarrow (IPort(p) \cup T)$ *is a set of dataflows among processes and objects,* $\Phi \subseteq (P \cup D) \times (P \cup D)$ *is a set of dataflows between processes and datastores, and* $\lambda : P \nrightarrow C$ *denotes the decomposition relations between processes in m and lower-level modules.*

## 3.3. Product Model Derivation Approach

Figure 4 shows the outline of our approach to deriving a specific product model through the integration of feature specifications. The derivation process can be divided into feature ordering and feature integration. During feature ordering, the features are selected from the feature model according to user needs. The order of features in a sequence may matter since feature composition is not generally commutative, and feature composition is not in every case associative [41]. We propose an *integration ordering* to achieve the ordering sequence. The integration ordering sorts each pair to be fused in order, and each pair is the integration of two features. During feature integration, *behavior preserving integration* is conducted for each feature pair in the order, which includes declaration integration, functional scenario path generation, and function integration based on path matching. We describe behavior in the form of a functional scenario path here, and the definition is given later. The declaration integration integrates variables and types in features according to integration rules. The functional scenario path connects the scenarios of each feature behavior. The function integration based on path matching matches related scenarios paths of feature pairs and integrates the feature specification. The key technologies *integration ordering* and *behavior preserving integration* in the above two steps will be described in detail below.

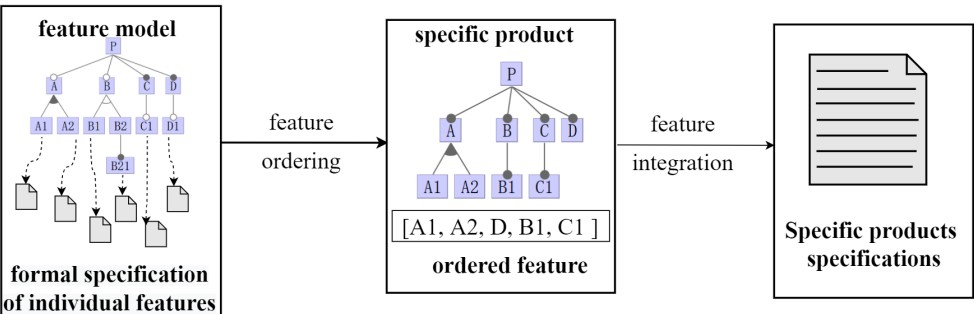

**Figure 4.** The outline of the product derivation approach.

### 3.3.1. Integration Ordering

The main goal of integration ordering is obtaining the sequence of pairs to be integrated. It includes the configuration of the feature model and the ordering of the selected features. Features are selected according to the user's needs, and the integration ordering is generated based on the product configuration.

In the feature model, basic features are composed by the superimposition of their tree structures. The main idea of ordering is that the more closely related features are integrated first. The bottom-level features of the model are treated as dispersed independent leaf nodes, and the closeness of the leaf node relationships is determined by the hierarchical structure of the model after product configuration. The feature model is deeply traversed to find all non-leaf nodes, and the leaf nodes are merged under the same parent node in pairs. The result is a merged leaf node. Then, the nearest neighbor search algorithm is used to find the nearest related leaf nodes, and the nearest related leaf nodes are merged until there is only one merged leaf node at the end.

Algorithm 1 shows the pseudo-code of the above process. In the algorithm, the input variable configuration is the user's operation for product configuration, which is unknown. The user selects features through configuration. The *juniorNodeSets* is a collection of all features in the feature model. For example, the leaf nodes A1, A2, B1, C1, and D in Figure 4 have a common root node, all the non-leaf nodes A, B, C, and P are found, and, when judging whether their children are leaf nodes or not, if they are, then merging all the leaf nodes under the same node, such as A1 and A2, the leaf nodes under B, C remain unchanged because there is only one leaf node. Then, the nearest nearby leaf nodes are found by the nearest neighbor search algorithm, merging them in turn, recursively, to obtain the final result as [A1, A2, D, B1, C1]. An arbitrary ordering sequence indicates the operation of integrating the result of every two features with the next feature. Each feature corresponds to a module in SOFL, so in this example, modules A1 and A2 are first integrated as a feature pair, denoted as $m_{f_{A1}} \cdot m_{f_{A2}}$, then, we merge D with the results of A1 and A2 integration, and obtain $m_{f_{A1}} \cdot m_{f_{A2}} \cdot m_{f_D}$; finally, the ordering is $m_{f_{A1}} \cdot m_{f_{A2}} \cdot m_{f_D} \cdot m_{f_{B1}} \cdot m_{f_{C1}}$.

### 3.3.2. Behavior Preserving Integration

Serving as the most basic integration operation for each feature pair, behavior preserving integration will be repeatedly applied for the selected features to be integrated in order. Regarding functional scenario path as the representation of the feature behavior, we perform the behavior preserving integration on this basis. Behavior preserving integrates every declaration and process that occurs in the module to cover all possible functional scenario paths, where a start conditional process is a conditional process whose input data stream is not the output of any other conditional process in the same module. It can also be a source (a conditional process without any input data stream). A termination conditional process is a conditional process, whose output data stream is not input to any other conditional process in the same module. It includes *declaration integration*, *functional scenario path generation*, and *function integration based on path matching*. The steps of the method are shown in Figure 5. The first step integrates all types of variable declarations according

to the integration rules. The second step derives all possible functional scenarios from the formal specification of the pair and then connects functional scenarios into scenario paths. Finally, the last step performs path matching according to the scenario paths, and the matching result guides the generation of integrated modules.

---

**Algorithm 1** Integration Ordering

**Input:** feature module $fm$, configuration $c$
**Output:** pair sequence $ps$

1   $selected feature \leftarrow c$;
2   $juniorNodeSets \leftarrow fm$;
3   // find all non-leaf nodes
4   $juniorTreeNode \leftarrow fm - selected feature$;
5   **for** $juniorTreeNode \in juniorNodeSets$ **do**
6     **while** $\exists Leaf Node \in juniorTreeNode$ **do**
7       // merge leaf nodes under the same parent node
8       $juniorTreeNode \leftarrow juniorTreeNode \cup leaf Node$;
9     **end**
10 **end**
11 $nearestNeighbor \leftarrow \varnothing$;
12 **while** $leaf NodeSets.count \geq 2$ **do**
13     // nearest neighbor search
14     **for** $\exists leaf TreeNode \in leaf NodeSets$ **do**
15       // find the leaf node with the closest relationship
16       $nearestNeighbor \leftarrow leaf TreeNode.findJuniorNodes.findLeaf Nodes$;
        // merge leaf nodes of nearest relatives
17       $leaf TreeNode \leftarrow leaf TreeNode \cup nearestNeighbor$;
18       $ps \leftarrow leaf NodeSets$;
19     **end**
20     **return** $ps$
21 **end**

---

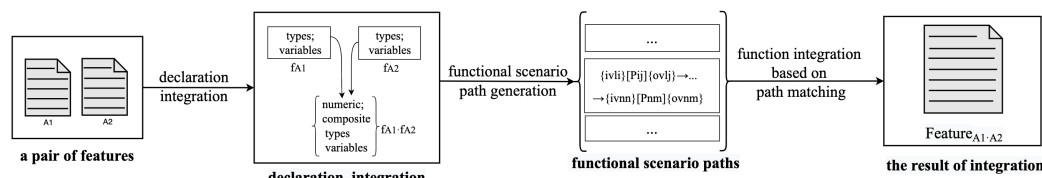

**Figure 5.** Integration of feature pairs.

**Declaration integration**. Declarations contain mainly types and variables. Variables can represent values of basic or compound types (e.g., sets, sequences, maps, or composite types). Two type declarations with the same name and base type are overwritten, and only one is left. Composite types have many data items. We focus on composite types with intersecting data items.

First, the declaration of the data items is analyzed in any two modules and then integrated according to the rules we give. Table 1 describes our proposed declaration integration rules to generate the same compound type variables as the original semantic from the relevant conditions involving the operation as mentioned above. Let s, s1, and s2 be any type, respectively. If the conditions in the table are satisfied, the actions will be executed. For example, the first and second rows of Table 1 indicate that if the domain/range of map s1 is satisfied by s2, then dom(s1) = s2 or rng(s1) = s2; the third row indicates if there is an intersection between map s1 and map s2, a new s will be generated, which is the union of s1 and s2; the fourth row indicates if there is an intersection between set s1 and set/com s2, the set s is the union (s1, s2); the fifth row indicates if sequence s1 and s2 partially overlap, the sequence s is conc(s1, s2); if set s1 and s2 partially overlap, then the composite s is the union(s1, s2).

**Table 1.** Rules for declaration integration based on composite type expressions.

| No. | Types | Conditions | Actions |
|---|---|---|---|
| 1 | Map | s1 = {a1→b1, a2→b2,..., an→bn} s2 = {b1, b2, . . . , bn} | rng(s1) = s2 |
| 2 | Map | s1 = {a1→b1, a2→b2,..., an→bn} s2 = {a1, a2, . . . , an} | dom(s1) = s2 |
| 3 | Map | s1 inter s2 not empty | comp(s) = union(s1, s2) |
| 4 | Set | s1 inter s2 not empty | set s = s1 union s2 |
| 5 | Sequence | s1 concatenated s2 | sequence s = conc(s1, s2), len(s) = len(s1) + len(s2) |
| 6 | Composite | s1 inter s2 not empty | comp(s) = s1 union s2 |

For type variables where two modules overlap exactly, we choose to retain one of them. If a type variable T has overlapping parts with two other type variables, T1 and T2, we integrate them in the order of the traversal, in turn. It is worth mentioning that most of the commonly used variable types are covered in our rules, and many other possibilities will be explored in the future.

**Functional scenario path generation.** The goal of the functional scenario path generation is to obtain a collection of functional scenario paths for each feature based on the functional scenario. The functional scenario path represents module behavior. A path indicates a set of functional scenarios, which describes how the final output data is produced by a sequence of processes based on the input data. A process may include several functional scenarios. The definition of a functional scenario is as follows:

**Definition 3.** *A **functional scenario** of a process p is a conjunction $preP_p \land C_i \land D_i$, where each $C_i (i \in 1, ..., n)$ is a predicate called a "guard condition" that contains no output variable and $\forall i, j \in 1, ..., n \cdot i \neq j \Rightarrow C_i \land C_j = false; D_i$ is called "defining condition" that contains at least one output variable but no guard condition in the post condition $(C_1 \land D_1) \lor (C_2 \land D_2) \lor ... \lor (C_n \land D_n)$.*

Each functional scenario $preP_p \land C_i \land D_i$ defines an independent behaviour, which it means that if $preP_p \land C_i$ is satisfied by the initial state (or the input data), the final state (or the output data) is defined by the defining condition, $D_i$. For example, in the simplified process of *Check_Pass* in Figure 3, according to the definition of functional scenarios, the three functional scenarios defined in the specification are listed as follows:

- $(\exists x \in Bank\_Account \cdot x.id = id \land x.password = pass\_no) \land sel = true \land acc1 = x$
- $(\exists x \in Bank\_Account \cdot x.id = id \land x.password = pass\_no) \land sel = false \land acc2 = x$
- $\neg(\exists x \in Bank\_Account \cdot x.id = id \land x.password = pass\_no) \land err\_msg =$ *"Your password or account number is incorrect"*

Since any postcondition can be transformed into the equivalent disjunctive normal form, simply treating the disjunctive clause in the disjunctive normal form of a postcondition as a functional scenario is not necessarily correct in some cases. For example, $x \leq 0 \land (y = x \lor y = -x) \lor x > 0 \land y = x/2$ is the postcondition of the operation, where $x$ is the input and $y$ is the output. It states that when $x \leq 0$, $y$ is defined as $x$ or $-x$. In this case, if we convert it to a disjunctive normal form $x \leq 0 \land y = x \lor x \leq 0 \land y = -x \lor x > 0 \land y = x/2$ and treat the two disjuncts separately, as in sentences $x \leq 0 \land y = x$ and $x \leq 0 \land y = -x$ as a separate functional scenario, we may not find a satisfactory answer in the analysis, then we can only judge that these two scenarios are classified as the same scenario (does not affect the integration result when the scenario path is integrated), or modify it when writing the product family's formal specification.

**Definition 4.** *Given a module m, a path of m is a sequence of processes $[p_i \cdots p_j]$, iff*

$$\exists_{p_i, \cdots, p_j \in P_m} \cdot (\exists_{n \in [i+1,j]} \cdot \nexists_{OPort(p_n) \cup T \times L \rightarrow IPort(p_i) \cup T}) \tag{1}$$

$$\exists_{p_i,\cdots,p_j\in P_m} \cdot \left(\exists_{m\in[i,j-1]} \cdot \nexists_{IPort(p_m)\cup T\times L\to OPort(p_j)\cup T}\right) \tag{2}$$

$$\forall_{oPort_j\in OPort(p_i),iPort_i\in IPort(p_{i+1})} \cdot oPort_j \cup iPort_i \neq \varnothing \tag{3}$$

The first condition means that there is no process in the path whose output is an input to $p_i$, and the second condition implies no process whose input is from $p_j$. The third condition indicates that each process should be connected to other processes with their input or output ports in the path. All paths connect processes $P_{ij}$ and $P_{nm}$, which are denoted by $Path(p_{ij}, p_{nm})$. The process $P_{nm}$ is obtained from the input of the *nth* port of a process $p$ in the module to the output of the *mth* port. A path can have one or more subpaths in the module.

**Definition 5.** *Given two paths, $Path(i,j)$ and $Path(m,n)$, where $Path(i,j)$ is a subpath of $Path(m,n)$, denoted as $Path(p_i, p_j) \subseteq Path(p_m, p_n)$, iff*

$$\exists_{x\in[m,n]} \cdot \exists_{OPort(p_x)\cup T\times L\to IPort(p_i)\cup T} \tag{4}$$

$$\exists_{y\in[m,n]} \cdot \exists_{Iport(p_j)\cup T\times L\to OPort(p_y)\cup T} \tag{5}$$

Each module corresponds to a set of functional scenario paths. All functional scenarios of a process $p$, denoted as $Fs_p$, have a functional scenario path, which is defined as follows:

**Definition 6.** *A **functional scenario path** is a sequence $[fs_1, \cdots, fs_n]$ in the module m, iff*

$$\exists_{[p_1,\cdots,p_n]\in \text{Path}_m(p_1,p_n)} \cdot \exists_{p_1,p_n\in P_m} \cdot \forall_{i\in[1,n]} \cdot \text{fs}_i \in \text{Fs}_{p_i} \tag{6}$$

The functional scenario path, denoted as $\text{FsPath}(p_{ij}, p_{nm})$, reflects the behavior of module $m_f$. The $m_f$ performs different functional scenario paths when receiving different data inputs. For simplicity, we only consider the postcondition in deriving functional scenarios discussed in this paper. All possible functional scenarios $fs_i (i \in \{1, 2, \cdots, n\})$ are derived from the formal specification of pairs based on the data dependency among operations $C_i \wedge D_i$, and a set of functional scenarios $\{\text{fs}_1, \text{fs}_2, \cdots, \text{fs}_n\}$ are obtained, where $fs_i$ is the related functional scenario of $p$. we use sequence $\{IV(\text{fs}_1)\}[\text{fs}_1]\{OV(\text{fs}_1)\} \to \{IV(\text{fs}_2)\}[\text{fs}_2]\{OV(\text{fs}_2)\} \to \cdots \to \{IV(\text{fs}_n)\}[\text{fs}_n]\{OV(\text{fs}_n)\}$ for the scenario path. Each $\{IV(\text{fs}_n)\}[\text{fs}_n]\{OV(\text{fs}_n)\}$ is a functional scenario. The input $IV(fs_1)$ of $fs_1$ comes from the objects outside the module $m$ and the set $OV(fs_1)$ is the outputs of $fs_1$. The process $P_{ij}$ in the scenario path receives the input data item $IV(fs_i)$ and produces a data item $OV(fs_i)$, where one or more variables in $IV(fs_i)$ are contained in the guard condition, and the variables in $OV(fs_i)$ are contained in a defining condition. The functional scenario path of each process $p$ within $m_f$ will be generated based on the functional scenarios. The process input and output variables $iv_{ij}, ov_{ij}$ can help to filter the functional scenarios of a process and obtain the relevant scenario paths. We implement the above process, as follows in Algorithm 2.

Here, we take the feature *ReserveforBus* and *ReserveforFlight* of TMS as an example. Figure 6 is the CDFD diagram of the formal specification of the feature. It can be observed from the figure that there are four processes in the module *ReserveforBus*: *Check_Bus*, *Check_Pass*, *Confirm_Bus*, and *Confirm_invoice* and four processes in the module *ReserveforFlight*. Each module has one or more functional scenario paths, as shown in Table 2. For example, there are four functional scenario paths for the feature *ReserveforBus*, of which the first scenario path *[reserve_for_bus_request, user_id]Check_Bus11[wrong_id]* is a functional scenario based on the process *Check_Bus*. The number 11 behind *Check_Bus* represents the port number, respectively; *reserve_for_bus_request, user_id* is from the first input port to enter the process *Check_Bus*, and the output is from the first port of the output after passing through the post-condition. The last three scenario paths are also similar.

---

**Algorithm 2** Feature Scenario Paths Generation

---

    **Input:** module $m_f$

    **Output:** *Path* /* the set of functional scenario paths         */

**1**  $postP_{m_f} \leftarrow P_1 \vee P_2 \cdots \vee Pn \ \ P_i(i \in \{1 \cdots n\})$;

**2**  **foreach** $P_t \leftarrow R_1 \wedge R_2 \cdots \wedge R_m (m \geq 1)$ **do**

**3**     |  $\{IO_1, IO_2\} \leftarrow iv_{ij} \subseteq IO_1, \ ov_{ij} \subseteq IO_2$

**4**  **end**

**5**  **foreach** $IO_k, k \in \{1, 2\}$ **do**

**6**     |  $C_1^t \leftarrow \wedge_{i \in s} R_i, s \in \{i \in [1 \cdots m] | R_i \in IO_1\}$;

**7**     |  $D_2^t \leftarrow \wedge_{i \in s} R_i, s \in \{i \in [1 \cdots m] | R_i \in IO_2\}$;

**8**  **end**

**9**  $fs_i \leftarrow C_1^t \wedge D_2^t$;

**10**  $Path \leftarrow \cup fs_i, i \in \{1 \cdots t\}$;

**11**  **return** $Path$;

---

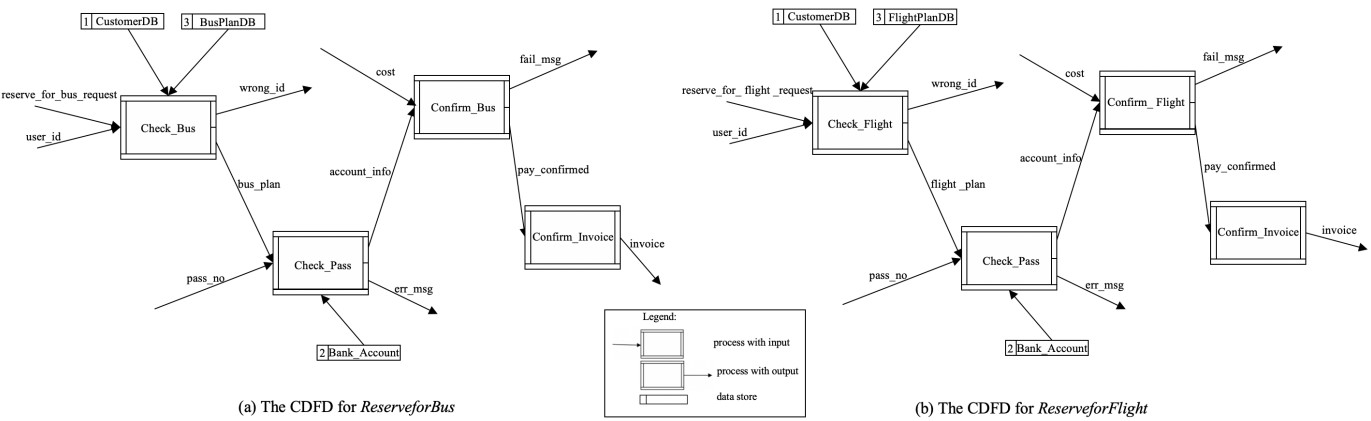

**Figure 6.** The CDFD for *ReserveforBus* and *ReserveforFlight*.

**Table 2.** The functional scenario paths involved in the selected modules.

| Feature/Module | Functional Scenario Paths |
|---|---|
| ReserveforBus | [reserve_for_bus_request,user_id]Check_Bus11[wrong_id] |
| | [reserve_for_bus_request,user_id]Check_Bus12[bus_plan]->[bus_plan,pass_no]Check_Pass11 [account_info]->[cost,account_info]Confirm_Bus11[failed_msg] |
| | [reserve_for_bus_request,user_id]Check_Bus12[bus_plan]->[bus_plan,pass_no]Check_Pass12 [err_msg] |
| | [reserve_for_bus_request,user_id]Check_Bus12[bus_plan]->[bus_plan,pass_no]Check_Pass11 [account_info]->[cost,account_info]Confirm_Bus12[pay_confirmed]->[pay_confirmed] Confirm_Invoice11[invoice] |
| ReserveforFlight | [reserve_for_filght_request,user_id]Check_Filght11[wrong_id] |
| | [reserve_for_filght_request,user_id]Check_Filght12[filght_plan]->[filght_plan,pass_no] Check_Pass11[account_info]->[cost,account_info]Confirm_Filght11[failed_msg] |
| | [reserve_for_filght_request,user_id]Check_Filght12[filght_plan]->[filght_plan,pass_no] Check_Pass11[account_info]->[cost,account_info]Confirm_Filght12[pay_confirmed]-> [pay_confirmed]Confirm_Invoice11[invoice] |
| | [reserve_for_filght_request,user_id]Check_Filght12[filght_plan]->[filght_plan,pass_no] Check_Pass12[err_msg] |

**Function integration based on path matching.** The goal of the integration is that all behaviors included in a single feature will be retained in the final feature's behaviors.

**Definition 7.** *Given two features $f_1$, $f_2$, let $S_{f_1 \cdot f_2} = (M_1, M_2, \cdots, M_n)$ denote a integrated feature specification, where each $M = (T_M, V_M, I_M, \mathrm{FS}_M)(i \in \{1, 2, \cdots n\})$ is a module defined in the specification, $T_M, V_M, I_M$ and $\mathrm{FS}_M$ are the set of all type declarations, state variable declarations, invariants, and functional scenarios, respectively. $f_1 \cdot f_2$ is said to be a behavior preserving integration, if*

$$\mathrm{FS}_{f_1 \cdot f_2} = \bigcup_{i=1}^{n} M_i \cdot \mathrm{FS}_M \tag{7}$$

$$T_{f_1 \cdot f_2} = \bigcup_{i=1}^{n} M_i \cdot T_M \tag{8}$$

$$V_{f_1 \cdot f_2} = \bigcup_{i=1}^{n} M_i \cdot V_M \tag{9}$$

$$I_{f_1 \cdot f_2} = \bigcup_{i=1}^{n} M_i \cdot I_M \tag{10}$$

$$\forall_{FsPath_{m_{f1}}, FsPath_{m_{f2}} \subseteq FsPath_{m_f}} \cdot \exists_{o \in OPort(P_{m_{f1}}) \wedge i \in IPort(P_{m_{f2}}) \wedge (i \wedge o \neq \varnothing)} \tag{11}$$

$$\forall_{FsPath_{m_{f1}}, FsPath_{m_{f2}} \subseteq FsPath_{m_f}} \cdot \exists_{i \in IPort(P_{m_{f1}}) \wedge o \in OPort(P_{m_{f2}}) \wedge (i \wedge o \neq \varnothing)} \tag{12}$$

*where $\mathrm{FS}_{f_1 \cdot f_2}$ are the set of all functional scenarios defined in the entire formal specification; $T_{f_1 \cdot f_2}$ and $V_{f_1 \cdot f_2}$ are the sets of all types and state variable declarations, respectively; and $I_{f_1 \cdot f_2}$ are the set of all invariants.*

The functional scenario is the basic unit in the formal specification for presenting the desired functions, and the process can be transformed into the conjunction or disjunction of functional scenarios. The module contains the set of all functional scenarios derived from the processes of f1 and f2. For each feature pair, feature integration can be conducted if its functional scenario path matches another feature. Function integration based on path matching aims to connect fragmented process paths into functional paths that can express independent operations. If the union of the scenario paths of modules provides multiple valid paths, a complete CDFD graph can be formed. It is called a matchable path. The matching of multiple process scenario paths in the feature can be regarded as the input and output of multiple sets aligned one by one. Therefore, the connection between paths is judged based on the data flow, and path matching is performed to form a complete scenario path operation.

Before the integration operation, the validity needs to be checked for each path. For two adjacent functional scenario paths, a post-condition that cannot be mapped to another pre-condition is an invalid path, that is $\forall_{FsPath_{m_{fi}}, FsPath_{m_{fj}} \subseteq FsPath_{m_f}} \cdot \exists_{\mathrm{postP}_{m_{fi}} \not\Rightarrow \mathrm{preP}_{m_{fj}}}$. Note that some checks, such as path checks related to data flow, may be conducted automatically by the tool, but having a manual check may reveal subtle defects that the machine cannot find. For example, in the ATM case [42], the user's withdrawal operation changes from *Receive_Command* to *Withdraw*, thus the data will not flow to the query balance. Such a mistake is hardly recognized by machines, but human beings can find it.

In terms of feature behavior, features will interact through both a database information exchange and process scenario path exchange. The database exchange means that the data stream of one operation will be read into the database, and the data stream of another operation will be written to the same database. This kind of information exchange is usually retained in the integrated features, and the module remains formally unchanged. In a simple case, the path matching needs to satisfy whether the connection between different port variables of the process can be matched. The conditions are as follows:

$$\exists_{FsPath_{m_{fi}} \in FsPath_{m_{f1}}, \, FsPath_{m_{fj}} \in FsPath_{m_{f2}}} \cdot FsPath_{m_{fi}} \neq FsPath_{m_{fj}}$$
$$\wedge (iPort(P_{m_{fi}}) = oPort(P_{m_{fj}}) \vee iPort(P_{m_{fj}}) = oPort(P_{m_{fi}})) \tag{13}$$

$$\exists_{FsPath_{m_{fi}} \in FsPath_{m_{f1}}, FsPath_{m_{fj}} \in FsPath_{m_{f2}}} \cdot FsPath_{m_{fi}} \neq FsPath_{m_{fj}}$$
$$\wedge (iPort(P_{m_{fi}}) \subseteq oPort(P_{m_{fj}}) \vee iPort(P_{m_{fj}}) \subseteq oPort(P_{m_{fi}})) \tag{14}$$

Condition 1 states that, for the functional scenario paths of different features $f_1$ and $f_2$, if a set of inputs and outputs are the same, the scenario paths are matchable. Condition 2 states that if the set of input variables of the functional scenario path of feature $f_1$ is a subset of the output variables of another $f_2$ or vice versa, it is intuitive to assume that they are related and can be matched.

Considering that there are multiple ways to connect the scenario, the connection of the complex scenario path can be divided into three cases, as shown in Figure 7. Figure 7a is the sequential path of scenario connections because the conditional process B will only be fired after the scenarios of processes A1, A2, ..., An are all satisfied. It can be formally expressed as $(pre_{A1} \wedge post_{A1}(v_1)) \wedge (pre_{A2} \wedge post_{A2}(v_2)) \wedge \cdots \wedge (pre_{An} \wedge post_{An}(v_n)) \Rightarrow pre_B$ where each $post_{Ai}(v_i)(i = 1, \cdots, n)$ is a sub-logical expression of the postcondition $post_{Ai}$, representing a component of one of the scenarios of process $A_i$. For process connections that select structures in Figure 7b, its connection is judged as $(pre \wedge post(v)) \wedge C(v) \Rightarrow pre_B$ or $(pre \wedge post(v)) \wedge \neg C(v) \Rightarrow pre_B$, where $pre$ is the precondition of the preceding condition process; $post(v)$ is the sub-logical expression of its postcondition, which contains variable $v$; and $pre_B$ is the precondition of the condition process B. For the process connection of the case structure in Figure 7c, its connection is judged as $pre \wedge post(v) \wedge C_i(v) \Rightarrow pre_{B_i}$, where $i = 1, \cdots, n$. If the variable $v$ satisfies the condition $C_i(v)$, the precondition that the scenario related to v is connected to the conditional process Bi needs to be guaranteed by the conjunction $pre \wedge post(v) \wedge C_i(v)$ so as to connect the scenario path correctly.

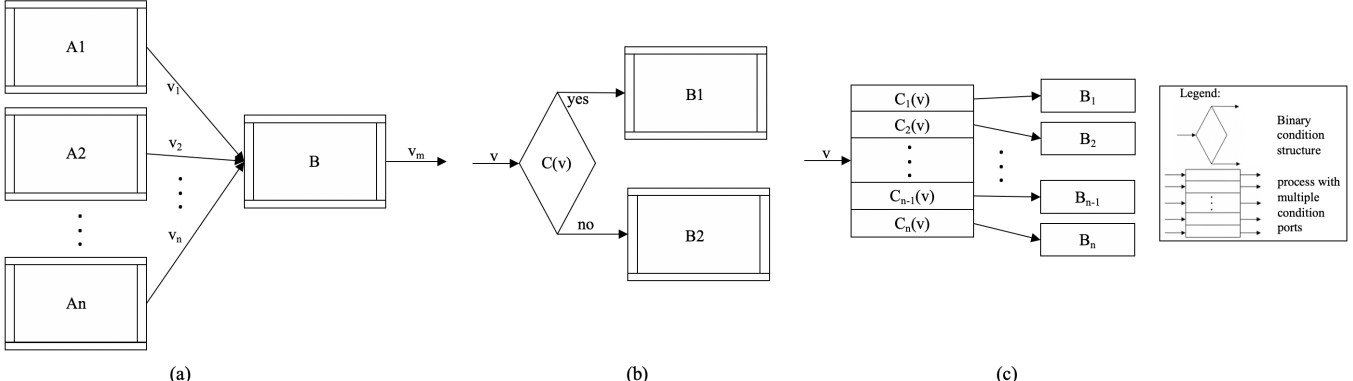

**Figure 7.** Different path connection structures.

After the path matching is completed, there will be some paths that may have data exchange but are not connected, and some feature fragments that have not been matched. For partially fragmented paths, our solution is to add templates (templates are empty processes), allowing users to fill in the syntax compliant input and output according to the functional scenarios. Here, the matching is only to find fragments that can be integrated and to detect whether the fragments of the specification can be integrated together. Integration merges the declarations of feature specifications according to the path, and the process fuses in turn according to the functional scenario path to form a larger feature specification. The above methods work well generally, but no guarantee can be given to ensure that a matchable path for integration will always be found or will always be efficiently found. If the function integration is unsuccessful, a manual check is required to determine whether there is a scenario path that satisfies the integration.

### 3.3.3. Validity Check

A validity check for a functional scenario path aims at checking the consistency between the integrated functional scenario path of the features and the desired scenario of independent features. Taking the strategy described in Definitions 8, the inspector

concentrates on the examination of whether or not every scenario defined in the integrated feature is implemented correctly by a single or set of paths in the path.

**Definition 8.** *Let $F_s = \{fs_1, fs_2, \cdots, fs_n\}$ be the set of all the functional scenarios defined in feature $f_1$ and $f_2$, and $FS_p = \{p_1, p_2, \cdots, p_n\}$ be the set of all the possible functional scenario paths of the integrated feature F. Then, F satisfies $f_1$ and $f_2$ if and only if there exists a mapping $M : F_s \to power(\mathrm{FS}_p)$ that satisfies the following condition:*

$$\forall_{fs \in F_s} \exists_{p \in power(\mathrm{FS}_p)} \cdot M(fs) = p \tag{15}$$

*where $power(\mathrm{FS}_p)$ denotes the power set of $\mathrm{FS}_p$, and $M(fs) = p$ $(p \in \mathrm{FS}_p)$ means that the set of the functional scenario paths p connects correctly the functional scenario $fs$.*

The connection of the scenarios of f1 and f2 to the paths can be automatically performed if all the variables and the logical expressions used in the feature specification are preserved directly in the integrated feature; otherwise, it can be performed manually with human support. The discovery of any inconsistency between the functional scenario in $f_1$ and $f_2$ and the corresponding scenario in *F* will indicate the existence of potential errors, and the nature of the inconsistency can be determined based on a rigorous path contrast analysis.

## 4. Supporting Tool and Case Study

### 4.1. Supporting Tool

We have developed a tool to support the product model derivation from a feature model and feature specification. The tool is developed using C# in the Visual Studio 2017 environment. The tool currently mainly offers three functions:

- Obtaining the feature integration order;
- Structuring and matching feature scenario paths;
- Integrating the formal specification of features.

The tool facilitates the user to select features, view the integration order, and allows feature function scenarios to be generated both automatically and manually before integrating feature specifications. Once the functional scenario path is valid, the tool will use the selected feature pair to generate integrated features by clicking the right button. The updated integrated feature specification can be obtained if the scenario path matching is complete.

Figure 8 shows a snapshot of the tool. The structure of the feature model in the current project is displayed in the upper left corner of our support tool, and its selected feature formal specification is given in the middle pane. When a feature is selected, its scenario path is automatically generated and rendered in the lower pane area. When you click the button in the upper right corner of the lower pane, you can operate on the scenario path or add a template. The integration process can be conducted automatically, or it can be conducted step-by-step with the user's operation.

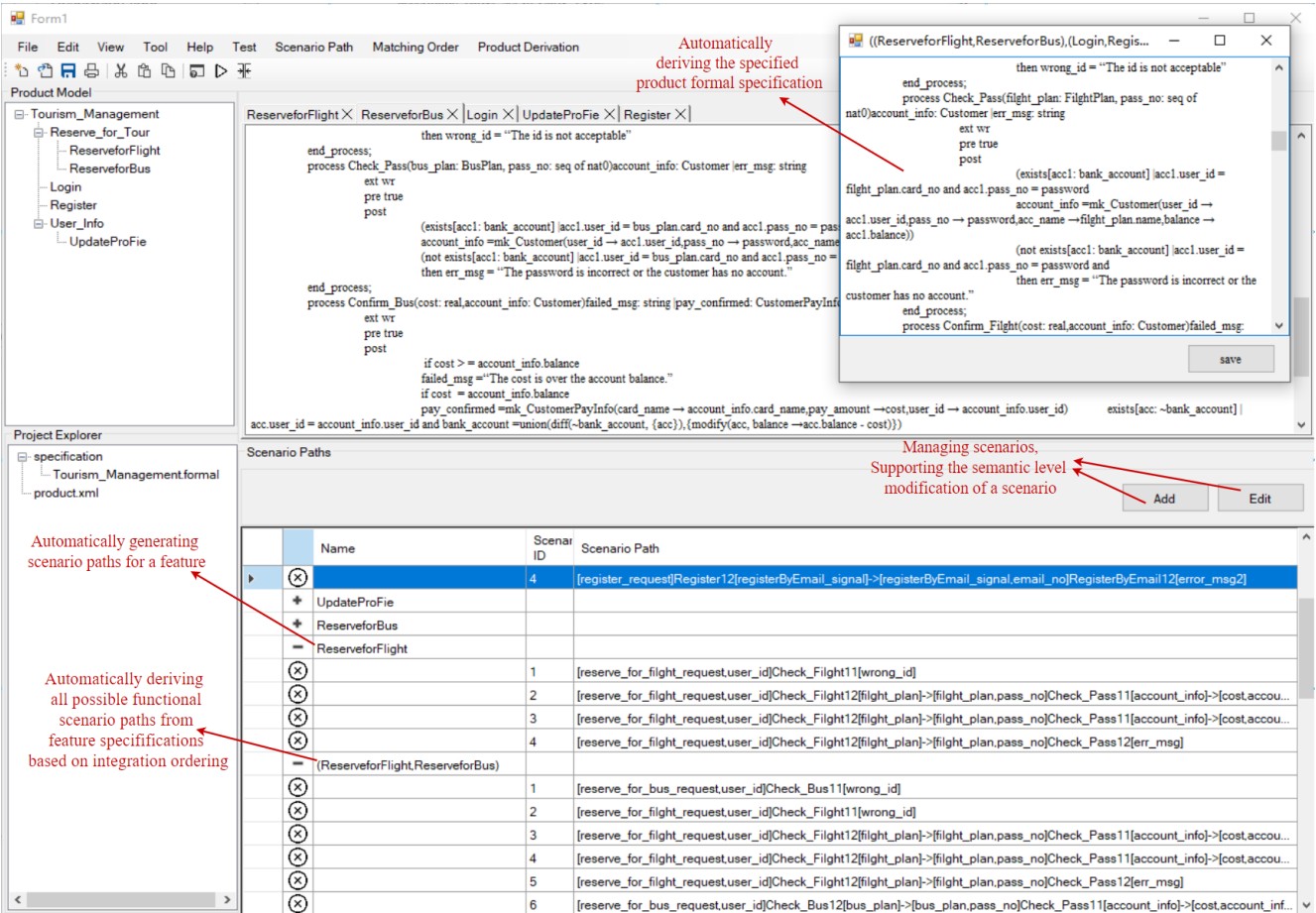

**Figure 8.** A snapshot of the tool.

*4.2. Case Study*

We have conducted case studies with four product family models and the corresponding formal specifications (including the Online Shopping System (OSS), Beverage Vending Machine System (BVMS), Hotel Room Reservation (HRR), and Tourism Management System (TMS)). In these case studies, we import the feature model and feature specification of the selected features and obtain the feature integration order through the support tool. The tool guides us to integrate the feature pair specification in order, and we judge the scenario path with the partial assistance of the tool and analyze whether the integration results are reasonable. If it meets our expectations, we perform the integration of the next feature pair, repeat the above steps, and obtain the formal specification of the specific product. Specifically, as shown in Table 3,

- The OSS product family has 14 features, the user selects 9 features, and these features contain 12 processes, of which the declaration integration used the rules 4, 5, and 6 in Table 1, and 5 paths are effectively matched. The final product's formal specification was as expected.
- The BVMS product family has 11 features, and the user selects 7 features, including 7 processes. Since the function is relatively simple, there is no declaration integration involved, and there are two valid paths matching obtain. The effect of the formalized specification of the product is not obvious.
- The HRR product family has 9 features, and the user selects 6 features, of which the rules 4 and 6 about Composite and Set collections in Table 1 are used in declaration integration, and 4 paths are effectively matched. The final product's formal specification is as expected.

- The TMS product family has 11 features. The user selects 5 of them, the declaration integration applies rules 4 and 6, and 4 paths are effectively matched. The final product's formal specification is as expected.

**Table 3.** The implementation details of four product family models' derivation.

| Product Family | Integration Order | Rules Used in Declaration Integration | The Number of Scenario Path | The Number of Scenario Paths Matched |
|---|---|---|---|---|
| OSS | [Login, Logout, FillOrderForm, SubmitOrderform, CheckOrderForm, SendInvoice, ReceiveInvoice, MakePayment, AcceptPayment] | Composite, set, sequence | 17 | 5 |
| BVMS | [Tea, Coffee, HotWater, Vegetable, Tomato, BalanceOut, Coinin] | \ | 7 | 2 |
| HRR | [CheckIn, CheckOut,ReserveRoom, ReserveCancel, Reservation, TelService] | Composite, set | 11 | 4 |
| TMS | [ReserveforFlight, ReserveforBus, Login, Register, UpdateProfile] | Composite, set , | 16 | 4 |

[1] Rules 4, 5, and 6 for set, sequence, and Composite collections in Table 1. [2] Rules 4 and 6 for set and Composite collections in Table 1.

We illustrate our approach in detail with a simplified product derivation process of the TMS product family (Figure 1) and its formal specification. The case study aims to illustrate how our approach integrates the feature specifications through a sequence of scenario paths and guides the user to construct a comprehensive description of a specific product.

According to the proposed approach, the feature model of the specific product and the formal specification of the product family is used as input. Since our method is applicable to the process from the product family to a specific product, the integration of feature specifications is usually conducted after the product configuration is completed. The selected features are shown in Figure 9.

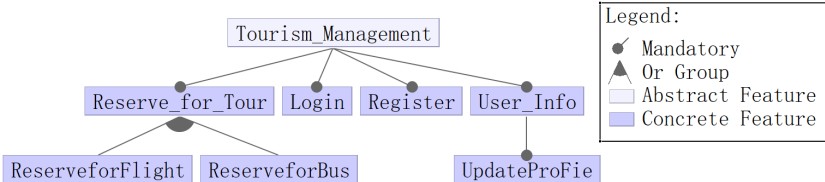

**Figure 9.** The selected features.

Based on the feature model structure and Integration Ordering algorithm, starting from the top root node, all nodes will be traversed to find non-leaf nodes. Then merge all the leaf nodes under the non-leaf nodes until the root node is merged. The resulting order is [ReserveforFlight, ReserveforBus, Login, Register, UpdateProfile].

We need to integrate the types and variables in the declaration first. Some feature specification declarations have great similarities. They do not involve the intersection of data and information, and the fusion results are not particularly obvious. Here, we list the declaration integration of two features, *Login* and *ReserveforBus*. The type variable that module *Login* contains has the basic information of the user (*id*, *pass_no*, *status*, *phone_no*, etc.), module *ReserveforBus* has type variables (*id*, *pass_no*, *card_name*, etc.). The variable *Customer* is a composite type in the two modules, including the common variables *user_id* and *pass_no*. The declaration integration needs to satisfy the composite type integration rules in Table 1, namely $comp(Customer_{Login \cdot ReserveforBus}) = Customer_{Login}$ union $Customer_{ReserveforBus}$. Only one is retained after integrating the same variables in the two modules *ReserveforBus* and *Login*. The variables in the two modules *ReserveforBus* and *Login* are user-defined variables. There is no information interaction,

so the two variables are directly retained in the integrated feature. The integration results are shown in Figure 10.

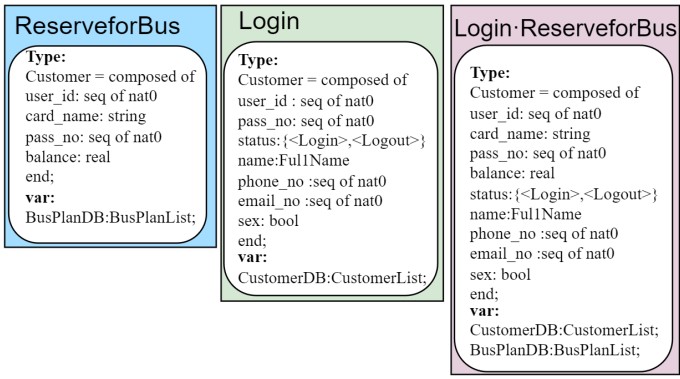

**Figure 10.** Declaration integration of feature pair.

The second step of feature pair integration is to fuse the process in the formal specification. The main judgment is based on the behavior of the feature pair, that is, whether the functional scenario path can be matched. We explained the process in detail by the integration of two features *ReserveforBus* and *ReserveforFlight*. Formal specifications for *ReserveforBus* and *ReserveforFlight*. are shown in Figure 11. We only focus on the functional integration based on scenario paths, so only the key parts are presented. *ReserveforBus* and *ReserveforFlight* have four processes, respectively, and their scenario paths are shown in Table 2. Intuitively, it can be found that they have similar operation processes, especially *Check_Pass* and *Confirm_invoice*. They have the same functional scenario at the module level, and the input and output port variables are the same, then the processes (*Check_Pass*, *Confirm_invoice*) can be merged.

The tool automatically finds the functional scenario of the feature by parsing the formal specification of the feature and guides the user to match the two features' functional scenario paths according to the ordering. The matched scenario paths are shown in Table 4.

**Table 4.** The integrated functional scenario paths of *ReserveforBus* and *ReserveforFlight*.

| Feature/Module | No. | Functional Scenario Path |
|---|---|---|
| reserveforBus,reserveforFlight | 1 | [reserve_for_bus_request,user_id]Check_Bus11[wrong_id] |
| | 2 | [reserve_for_filght_request,user_id]Check_Filght11[wrong_id] |
| | 3 | [reserve_for_filght_request,user_id]Check_Filght12[filght_plan]-> [filght_plan,pass_no] **Check_Pass23[ account_info2]**-> [cost, **account_info2**]Confirm_Filght11[failed_msg] |
| | 4 | [reserve_for_filght_request,user_id]Check_Filght12[filght_plan]-> [filght_plan,pass_no] **Check_Pass23[ account_info2]**-> [cost, **account_info2**]Confirm_Filght12[ **pay_confirmed2**]-> [ **pay_confirmed2**] **Confirm_invoice11**[invoice] |
| | 5 | [reserve_for_filght_request,user_id]Check_Filght12[filght_plan]-> [filght_plan,pass_no]Check_Pass12[err_msg] |
| | 6 | [reserve_for_bus_request,user_id]Check_Bus12[bus_plan]-> [bus_plan,pass_no] **Check_Pass11[ account_info1]**-> [cost, **account_info1**]Confirm_Bus11[failed_msg] |
| | 7 | [reserve_for_bus_request,user_id]Check_Bus12[bus_plan]-> [bus_plan,pass_no] **Check_Pass11[ account_info1]**-> [cost, **account_info1**]Confirm_Bus12[ **pay_confirmed1**]-> [ **pay_confirmed1**] **Confirm_invoice11**[invoice] |
| | 8 | [reserve_for_bus_request,user_id]Check_Bus12[bus_plan]-> [bus_plan,pass_no]Check_Pass12[err_msg] |

There are eight scenario paths after the integration, and the process *Check_Pass* and *Confirm_invoice* marked in bold black have changed compared with those before the integration. Process *Check_Pass* used to have one input and two outputs, but now it has two inputs and three outputs. Process *Confirm_invoice* used to have one input but now has two inputs. The corresponding integrated CDFD is shown in Figure 12. The formal specification of the integrated feature pair is shown in Figure 13. After integration, the user can add templates to ensure complete functional operation for *ReserveforBus* and *ReserveforFlight*. As the final step of product derivation, the tool repeatedly guides the user through the integration of feature pairs until the final full product formal specification is obtained.

```
module ReserveforFlight;
    consts
        Today = Computer.date;
    types ···
    var ···
    inv
    process Check_Filght(reserve_for_filght_request:signal, user_id:nat0)wrong_id: string | filght_plan: FilghtPlan ···
    end_process;
    process Check_Pass(filght_plan: FilghtPlan, pass_no: seq of nat0)account_info: Customer |err_msg: string
        ext wr
        pre true
        post
            (exists[acc1: Bank_Account] |acc1.user_id = filght_plan.card_no and acc1.pass_no = password
            account_info =mk_Customer(user_id → acc1.user_id,pass_no → password,acc_name →filght_plan.name,balance → acc1.balance)
            (not exists[acc1: Bank_Account] |acc1.user_id = filght_plan.card_no and acc1.pass_no = password and
            then err_msg = "The password is incorrect or the customer has no account."
    end_process;
    process Confirm_Filght(cost: real,account_info: Customer)failed_msg: string |pay_confirmed: CustomerPayInfo ···
    end_process;
    process Confirm_invoice(pay_confirmed: CustomerPayInfo)invoice: Invoice
        ext wr
        pre true
        post
            invoice =mk_Invoice(name →pay_confirmed.card_name,paid_total →pay_confirmed.pay_amount,date → Today)
    end_process;
end_module;

module ReserveforBus;
    consts
        Today = Computer.date;
    types ···
    var ···
    inv
    process Check_Bus(reserve_for_bus_request:signal, user_id:nat0)wrong_id: string | bus_plan: BusPlan ···
    end_process;
    process Check_Pass(bus_plan: BusPlan, pass_no: seq of nat0)account_info: Customer |err_msg: string
        ext wr
        pre true
        post
            (exists[acc1: Bank_Account] |acc1.user_id = bus_plan.card_no and acc1.pass_no = password
            account_info =mk_Customer(user_id → acc1.user_id,pass_no → password,acc_name →bus_plan.name,balance → acc1.balance))
            (not exists[acc1: Bank_Account] |acc1.user_id = bus_plan.card_no and acc1.pass_no = password and
            then err_msg = "The password is incorrect or the customer has no account."
    end_process;
    process Confirm_Bus(cost: real,account_info: Customer)failed_msg: string |pay_confirmed: CustomerPayInfo ···
    end_process;
    process Confirm_invoice(pay_confirmed: CustomerPayInfo)invoice: Invoice
        ext wr
        pre true
        post
            invoice =mk_Invoice(name →pay_confirmed.card_name,paid_total →pay_confirmed.pay_amount,date → Today)
    end_process;
end_module;
```

**Figure 11.** Formal specification of *ReserveforBus* and *ReserveforFlight*.

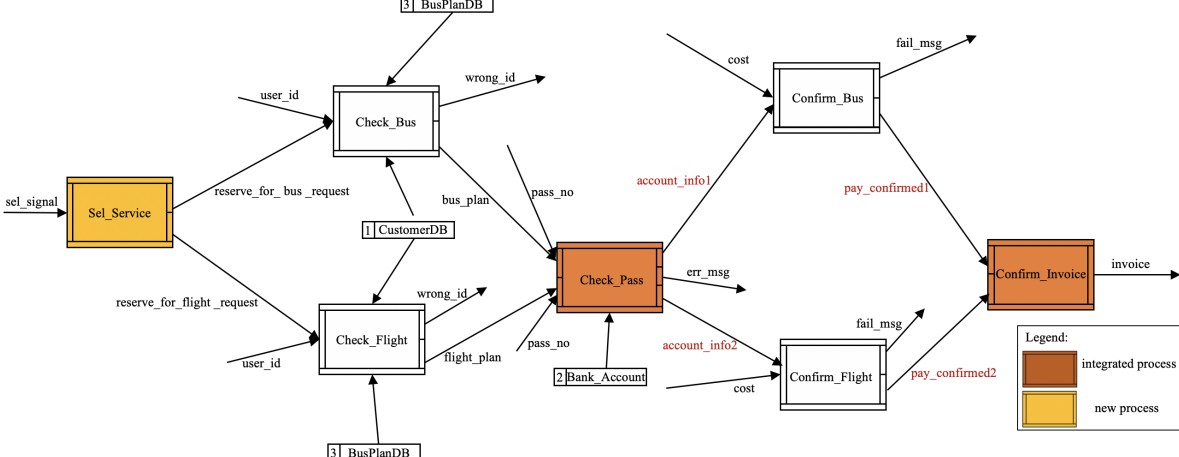

**Figure 12.** CDFD diagram of integrating features.

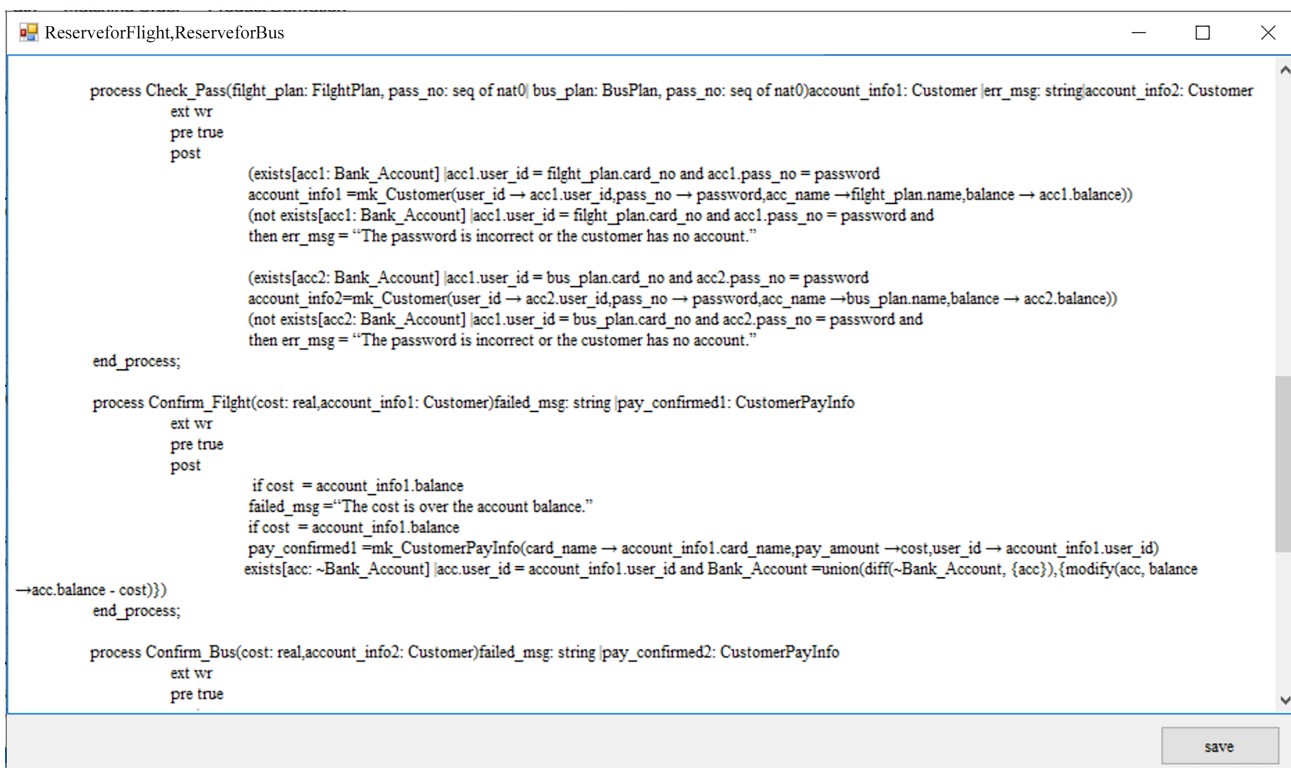

**Figure 13.** Formal specification of integrated feature pair.

## 5. Discussion

Through our experience with several case studies that we have conducted, we have found several benefits of our approach and, at the same time, found some limitations of our approach.

### 5.1. The Advantage of our Method

Our method has effectiveness in product derivation. Supported by the tool, our method is not limited to product derivation at the feature level. It can fully identify and understand the internal behavior of each feature, and the obtained product description is accurate. In addition, the requirements for operators are not high, and it only needs to understand the basic SOFL syntax. For managers and practitioners, our method can accurately grasp user needs through feature behaviors and provide a product derivation guideline; for academics, it can provide a new way of thinking about how to describe products accurately in terms of product derivation.

The supporting tool can capture all functional scenarios of the feature specification and provide reasonable integrating results for the user to confirm. In particular, if feature behaviors are unreasonable, we can effectively guide users to perform feature pair integration operations by adjusting and deleting scenario paths or adding templates, which shows the flexibility of our method. Therefore, we can remove unreasonable scenarios and add template operations to fit the needs of real-world users, effectively guiding the generation of product models.

### 5.2. The Limitations of our Method

Applying the tool-supported approach in the case study, we have also found some limitations. The first is to check the validity of the scenario path matching. Although paths can exhibit feature behavior, determining whether the integration results meet user intent still depends, to a certain extent, on human experience and understanding of the requirements and domain knowledge. The second major limitation is that the variability of features is not well represented. The main reason is that we take the selected features and

formal specifications as input in the product configuration phase, assuming that the user selects the functions required by the final product.

Despite these limitations, through the case study, we believe that the methodology can help researchers facilitate the product derivation process in practice.

## 6. Conclusions

In this paper, we propose a product derivation method that can computer-aid the construction of formal product models from feature models and feature specifications, filling the method gap of formal models from product families to specific products. Formal specifications for building a product are critical for stages such as coding, validating, and maintaining subsequent products. Our approach builds on product families to generate specific products by integrating formal specifications for each feature based on feature models and formal specifications. The method further proposes a behavioral preservation mechanism based on functional scenarios of feature formal specification, which ensures the consistency of formal specification before and after integration and ensures the integrity of the final product model.

Due to the well-defined formal syntax and semantics of the method, we developed a supporting tool for our method with multiple integrated functions to guide the product model-building process and reduce the burden of product derivation. Furthermore, we apply the method through the tool to a case study in which the integrated results improve the quality of derivative products and illustrate the feasibility and effectiveness of our method.

In the future, we will continue to extend our tool with more capabilities for product derivation. We will also focus on the application in complex/large applications and the verification of the consistency in the behavior preserving integration (e.g., checking that feature interconnections in the derived product are correct) and developing criteria or algorithms that support integration in the presence of complex path mismatches, and at the same time improve our support tools to make them more user-friendly.

**Author Contributions:** Formal analysis, X.W., W.W. and H.L.; writing—original draft, W.W.; writing—review, X.W. All authors have read and agreed to the published version of the manuscript.

**Funding:** This research was funded by the NSFCs of China (No. 61872144 and No. 61902234), National Social Science Foundation (No. 17AZX003) and Key Projects of Philosophy and Social Sciences Research, Ministry of Education (No. 18JZD013).

**Institutional Review Board Statement:** Not applicable.

**Informed Consent Statement:** Not applicable.

**Data Availability Statement:** Not applicable; the study does not report any data.

**Conflicts of Interest:** The authors declare no conflict of interest.

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
