# Peer review of "Product Model Derivation from Feature Model and Formal Specification"

_applsci, doi:10.3390/app12126241_

Round 1

Reviewer 1 Report

The paper is good and should be publication-ready after a suitable revision. 

  The contribution of the research should be spelled out more effectively.  The flow of the paper needs some work. The literature review can be supplemented with more papers.  Use of language needs to be improved.  The research question needs to be analyzed better.  Tables and diagrams need to be explained effectively and revised so that they are easy to read, and sources need to be included.  The conclusion needs to be a lot more than two paragraphs, here the work must be analyzed, and its contribution and importance spelled out.  Equations must be numbered.     All the best with the revision, I am looking forward to the next version. 

Reviewer 2 Report

1.      In the abstract, it is suggested to clearly describe how a case study is conducted in order to illustrate the method’s effectiveness and what the research findings are.

2.      It is also suggested to add the implication to this paper. How does this proposed product derivation approach contribute to the body of knowledge?

3.      The authors need to justify this statement with the evidence, “However, they are not suitable for direct application to product model construction because they lack a comprehensive and systematic perspective to consider the specific functions within features and the relationship between features.”

4.      In this sentence, “based on previous work, we propose a method for product derivation at the requirements level that integrates the feature specification, which focuses on feature specification rather than code implementation”. What is the previous work? How the authors propose a method based on the previous work? Please make it clear.

5.      Please elaborate more on “features A and B” are feature pairs.

6.      For this sentence, “However, we use Structured Object Oriented Formal Language (SOFL) because we are familiar with SOFL”. How can the authors choose SOFL because they are familiar with SOFL? Are there any other reasons?

7.      The authors stated that …several case studies we have conducted… How many case studies are there? How are these case studies conducted? The authors need to make it clear and in details.

8.      For the advantages, the authors need to explain and elaborate more how their method can accurately grasp user needs through feature behaviors and provide a product derivation guideline.

9.      More recent references (within 5 years) should be cited and added.

Reviewer 3 Report

Paper 1749535 review to Applied Sciences – Product Model Derivation from Feature Model and Feature Specification

All issues raised in this review can be considered to be minor reviews.

General considerations

The subject of Product Model Derivation is very relevant and current, being certainly a support method, which makes the design and development phases of new products more flexible, as well as the implementation of improvements in existing products, among other phases that precede production, that is, the planning of works. The article is very well prepared, with a high quality content approach, which is explained very clearly and reasoned with a case study with the fundamentals and ideas very well-articulated.

Article structure

The structure of the article is well elaborated, with no flaws detected in the numbering of the sections (chapters) presented, as well as in the subtitle numeration of figures and tables. But a lack of numbering of the equations was detected, therefore, the authors must number them.

Title, Abstract and Keywords

·         The title is appealing to readers. But the word “Feature” appears twice, so, the second one should be removed or be replaced by the word "Formal" to comply with the Abstract.

·         The abstract is well constructed. The main research question, the objectives and the development of the theme are clearly pointed out.

·         The keywords are adequate.

Affiliation

Affiliations seem to be correct in terms of numeration.

Figures and tables

The figures, tables and equations are all well designed. But there are some issues that must be corrected, namely:

·         In figures 6 and 11, several arrows are over words, so the positioning of arrows or words must be corrected;

·         The legends of figures 6 and 7 are illegible, therefore, the authors should enlarge them.

Grammar, spelling and syntax issues

The whole article it's well written in terms of grammar and spelling. But there were identified some aspects that should be improved/corrected, namely in the line(s):

·         69, the three words of the full meaning of the acronym DFD must begin with capital letters;

·         71, the four words of the full meaning of the acronym CDFD must begin with capital letters;

·         94, the acronym UML should be accompanied by its full meaning, but only the first time it is used;

·         119, the acronym SOFL should perhaps be accompanied by its full meaning, but only the first time it is used, despite being the focus of approach of section 3.2;

Semantic and technical issues

The entire article is very well explained. The issues are explained very clearly and the concepts and ideas are very well articulated between themselves.

References

The list of references is very well prepared, the number of references is appropriate to the depth of the theme's approach in the article and the text is very well referenced. The references are strong and most of them recent, in the scope of this investigation.

Reviewer 4 Report

The paper proposes an approach supported by a tool to obtain a formal specification of a specific product based on the feature model and formal specification. The work is interesting. However, the presentation must be improved since novelty, and scientific contribution are not clear.

1) Scientific contribution of this paper is not clear. It seems that the tool has been already described in [15]. Hence, the proposed approach is not new either. The authors must better describe how this work improves the authors’ previous work on this topic.

2) The related work section must be improved. The current description is shallow, and the differences between the proposed approach and similar works are not detailed enough. Furthermore, a broader context is needed. The authors might mention production modelling languages, model-driven round-trip engineering where different configurations can be obtained, model evolution when requirements are changed, model variability management, etc.

3) Structured Object-Oriented Formal Language (SOFL) and Conditional Data Flow Diagrams (CDFD) are outdated. The authors must show (supported by references) that these specifications are nowadays still used in the industry.

4) I am missing the research questions.

5) Experimental part is weak and must be improved.

6) Most of the references are rather old. Are the authors aware of recent advances in this field?

7) Typos:

and process(A11,   ->  and process (A11,

 baesd on

->   // Fig. 5

based on

 TMS product family(Figure 1)  ->  TMS product family (Figure 1)

 References used in this review:

======================

Vještica et al. 2021: Multi-level production process modeling language. Journal of Computer Languages, Volume 66, October 2021, 101053

Marah et al. 2021: Model-driven round-trip engineering for TinyOS-based WSN applications, Journal of Computer Languages, Volume 65, August 2021, 101051

Feichtinger et al. 2021: Guiding feature model evolution by lifting code-level dependencies. Journal of Computer Languages, Volume 63, April 2021, 101034

Nieke et al. 2021: Augmenting metamodels with seamless support for planning, tracking, and slicing model evolution timelines. Journal of Computer Languages, Volume 63, April 2021, 101031

Tërnava and Collet, 2021: A framework for managing the imperfect modularity of variability implementations. Journal of Computer Languages, Volume 61, December 2020, 100998

Fronchetti et al. 2022: Language impact on productivity for industrial end users: A case study from Programmable Logic Controllers. Journal of Computer Languages, Volume 69, April 2022, 101087

Round 2

Reviewer 4 Report

My comments have been addressed and the paper can be accepted now.

This manuscript is a resubmission of an earlier submission. The following is a list of the peer review reports and author responses from that submission.

Round 1

Reviewer 1 Report

Paper is well written and organized. I find some merit and it could be useful for academics and practitioners. I will recommend some minor changes.

  1. A Discussion section is necessary. The contribution of the author’s approach to the literature is not highlighted. The literature review needs to be integrated with the claims that the author make in order to show the importance of its contribution.

In the discussion section include the summary of the findings and highlight the he contribution of your study to the literature. Authors should compare their findings with other works in the literature, standing out what their contribution to the State of the art is, and if the findings fit with what was expected 

Overall, try to provide sufficient validation regarding the novelty of this research along with beneficial.

In addition, implications for managers and practitioners, and academics, should be highlighted in the discussion section.

  1. Authors should improve the conclusion section. Begin with a brief summary of paper motivations, objectives and findings. Finally, include your limitations and future research

Reviewer 2 Report

The paper is concerned with deriving a product model based on features model and formal specifications, aiming the integration of formal specification for each pair of features. In this knowledge domain the paper gives a good contribution to enhance the corresponding state-of-art since the proposed approach is able to cope with users’ requirements providing accurate product models and getting a formal product specification. The paper is well written and structured making easy to follow the authors’ ideas. A comprehensive description of the theoretical stuff underlying the authors’ proposed approach is also included. Furthermore, a suitable survey of published material related with the paper’s topic is provided and the discussion of the results obtained in these publications is addressed. The robustness of the proposed approach is assessed considering a case study. Thus, in the reviewer opinion the paper is recommendable for publication subject to a minor change, according to the following comment:

  • Figure 9 may have added value to reader has an idea about the tool obtained to implement the authors proposed approach. However, in the current form it is meaningless since no one can read it having only a perception that is a computer screenshot. Thus, it is recommended improvements in the mentioned figure, making it readable.

Reviewer 3 Report

The goal of this submission is to present a formal, logic based, product derivation method to obtain formal feature specifications. The method utilises a feature model and the SOFL language. 

However, although aiming at mathematical precision and formality, the authors use key notion such as "matching", "integration ordering", "process"  and "behavior-preserved" very informally, i.e., without proper definitions. They use also imprecise terms such as "Boolean logic". Even worse, the authors do not say what SOFL really is; which part of it they are using etc. I guess that they intend to use a kind of first order predicate calculus, but it was nowhere clearly stated. 

This makes the understanding of the paper very hard,  and even impossible at many places. 

My understanding of the authors' intention is that they try to compose specifications formulated in a first order predicate logic using a matching algorithm concerning variables occurring in the corresponding formulas. In this way, the post-conditions of one specification are composed with the pre-conditions of another. It seems that they call such compositions processes, however this was not clearly stated. 

The problem with such approach is that variable matching does not suffice here. The key problem is  that the logical consequence is not a decidable relation. In order compose one post-condition with other pre-condition it is necessary to know if the pre-condition is satisfied. In general, this requires proving that the post-condition implies the pre-condition. Otherwise, what is the sense of the specification composition?

The paper contains many unclear fragments and English requires improvement mostly because of the lack of clarity. 

In my opinion, even if the authors had sound concepts, the paper requires a lot of work concerning presented ideas and their formulation. 

I attached a PDF-file with detailed remarks for the authors. 

Reviewer 4 Report

Authors propose a product derivation approach to generate specific products by integrating the formal specification of each feature according to the requirements based on the feature model and feature specification. The idea is interesting and useful in practical scenario. Author also developed a tool with several integration capabilities.

My suggestions or findings are as under:

  • The practical aspect in Realtime scenario need to be mentioned in details. E.g., Why, When and How the proposed framework is useful.
  • What about the similar existing approaches? Do some comparison in order to indicate supremacy of the existing work.
  • The framework/methodology discussed so much technicality with equations and definitions. For better understanding , authors needs to discuss in details regarding the actual adopted methodology.
  • The details of the developed Tool are very limited. A single and very small example is carried out in the given case study. How your approach will behave for complex/large applications. Discuss toll in details so that the perspectives researcher can be able to experiment and also replicate the existing case study.   

I would like to see the above mentioned points in the updated manuscript.

Round 2

Reviewer 3 Report

The new version of the submitted paper contains a number of improvements addressing some of the problems I have mentioned in my previous review.

However, the main problems were not addressed and the changes are rather minor.

First of all, it is still unclear to me what is the goal of this paper and how the authors are going to achieve it. The notion of behaviour preserving integration seems to play a crucial role, as it is mentioned several times starting from the abstract and the paper's narrative is built around it. Unfortunately, the notions of behaviour and behaviour preservation are not defined, nor even explained in a clear way. Thy are far from being trivial and, thus, require a clear definition. At the end of the paper, the authors define the notion of "behaviour preserving integration", but they do not explain it properly. The notion seems to be related rather with composition than with the notion of behaviour as it is usually understood, i.e., it does not concern change. Initially, I expected that the authors would prove or at least demonstrate that the composition indeed preserves behaviour in a certain way, but there is not even mention of that. The definition seems to address rather the way the parts are composed but not the behaviour of components and of the whole.  

In my previous review, I pointed out that there is a fundamental problem with proving properties of behaviour composition if it is specified in the first order logic because in this case the relation of logical consequence is not decidable. The authors use pre- and post-conditions defined in SOFL. Although they still did not present the language properly, I guess that it is based on the first order logic. In consequence, the consequence relation is not decidable.

The authors should define SOFL precisely because it is not clear from the paper what kind of language it is, even if it contains predicates. They mention it in their reply but not in the paper.

The authors write in their reply to one of my objections:

 we also added a validation check when we actually did integration. After the matching is successful, we also need to do an integration check, which will take into account pre-and post-conditions. The valid matching results will be integrated. We have modified lines 348-350 of the revised manuscript to make the expression clearer. However, we can only solve some simple cases. We still need to manually check the validity when we cannot judge the more complex pre-and post-conditions. We will improve this aspect in the future.

The above mentioned “validation check” is not mentioned in the paper; neither the “integration check”.  The authors mention in the paper “validity check” but do not explain what they mean. If the validity check has anything to do with logical reasoning in FOL, then it cannot be done “manually”. Theorem proving is intellectually very demanding and anything but manual. Existing, the so called, theorem provers can prove by their own only very simple properties and for more complex prove need a very skilful guidance of a human being.

The writing style is unclear and English language is still in need of improvement.

To sum up, in my opinion a lot of work is needed to explain the authors ideas properly, to define them formally and to validate them appropriately.

Reviewer 4 Report

Extend required pending work as future directions.
